# Zoonotic hepatitis E virus spreads through environmental routes in pig herds – A phylodynamic analysis

Marina Meester[1]*, Cecilia Valenzuela Agüí[2,3◉], Tijs J. Tobias[1,4◉],
Renate W. Hakze van der Honing[5], Claire Guinat[6], Martijn Bouwknegt[7],
Louis du Plessis[2,3], Egil A.J. Fischer[1], Mirlin Spaninks[1], Tanja Stadler[2,3],
Wim H.M. van der Poel[5], Arjan Stegeman[1]

**1** Department of Population Health Sciences, Faculty of Veterinary Medicine, Utrecht University, Utrecht, the Netherlands, **2** Department of Biosystems Science and Engineering, ETH Zürich, Basel, Switzerland, **3** Swiss Institute of Bioinformatics, Lausanne, Switzerland, **4** Royal GD, Deventer, the Netherlands, **5** Wageningen University and Research, Lelystad, the Netherlands, **6** IHAP, Université de Toulouse, INRAE, ENVT, Toulouse, France, **7** Vion Food Group, Boxtel, the Netherlands

◉ These authors are equal contributions.
* M.meester@uu.nl

## Abstract

Worldwide, many pig farms are affected by hepatitis E virus (HEV) genotype 3, a zoonotic virus that causes hepatitis in humans. People can become infected after eating contaminated pork, making HEV control in pig farms crucial for public health. However, knowledge of HEV transmission dynamics and control options within farms is limited. Our findings reveal that HEV persists in the farm environment, enabling transmission between pigs separated in space and time. We investigated HEV transmission on two Dutch finishing farms for nine months in 2022. In both farms, samples from three compartments (confined rooms), holding 12 pens with pigs each, were collected and tested weekly across three batches (consecutively housed groups of pigs). Additionally, at least one sample per HEV-positive pen was sequenced per batch, retrieving 89 near-complete sequences. We integrated epidemiological data on duration and timing of infection with phylogenetic data to quantify transmission. We observed phylogenetic clustering of pens per compartment in both farms. In farm A, some sequences from different compartments and different batches also clustered, suggesting transmission between pigs housed separately. In farm B, only one compartment became HEV-positive during one batch. Within that compartment, between-pen transmission was efficient, with an effective reproduction number ($R_e$) of 3.6 (95% HPD interval 1.3–6.7). The other compartments and batch may have remained HEV-negative thanks to stringent biosecurity measures applied on that farm. In farm A, the $R_e$'s for transmission between pens within and across compartments were not significantly above 1, yet all sampled pens became positive in all batches. A combination of transmission routes, in conjunction with persistence of

**Data availability statement:** All relevant data on infection dynamics, sampling results, and input for phylodynamic models, are within the manuscript and its supporting information files. All sequence data can be accessed via the HEVnet gobal repository for hepatitis E virus sequence data (https://www.rivm.nl/en/hevnet). The login link https://secure.rivm.nl/mpf/database/hev, the Database IDs are: 11390 – 11477.

**Funding:** This work was part of the Dutch public-private partnership project "HEVentie: hepatitis E virus intervention in primary pig production". HEVentie received financial support for staff, materials and analysis of the Topsector AgriFood (TKI AF 18119). In addition, the work was supported by the European Union's Horizon 2020 Research and Innovation Programme under grant agreement No. 773830: One Health European Joint Programme (BIOPIGEE: Biosecurity Practices for Pig Farming Across Europe. The funders had no role in study design, data collection and analysis, decision to publish, or preparation of the manuscript.

**Competing interests:** The authors have declared that no competing interests exist.

HEV in the environment, is required to explain why all pens tested positive. These findings show not only how HEV effectively spreads without pigs sharing housing, yet also that reduction of HEV's zoonotic risk may be achieved by improved biosecurity within farms.

## Author summary

Hepatitis E virus is the dominant cause of acute viral liver infections globally. In industrialized countries, human infection with the virus is mainly caused by consumption of raw or undercooked pork. This route of infection could be controlled by preventing hepatitis E virus from spreading between pigs, so no new pigs become infectious to humans. Currently, we do not know how to prevent spread of hepatitis E virus in farms, so most pigs become infected. By following up several groups of pigs for nine months on two farms, we could determine how the virus is spreading. Our findings reveal that hepatitis E virus can persist in the farm environment, for instance in housing that has not been cleaned entirely, or on materials that are shared between groups of pigs. This way, the virus has been able to spread between groups of pigs that are not in direct contact with each other. By implementing stricter cleaning procedures that lower environmental persistence of hepatitis E virus, the indirect spread between groups of pigs could be drastically reduced. Our results are shared with farmers and veterinarians, to help them control hepatitis E virus in their farms, and therewith reduce the risk of human infection.

## Introduction

Hepatitis E virus (HEV) is a positive-sense RNA virus of the family of *Hepeviridae,* species *Paslahepevirus balayani*, and the primary cause of acute viral hepatitis worldwide, with an estimated 1 in 8 people having been infected with this virus [1]. In industrialized countries, HEV genotype 3 is the predominant genotype, and in Europe, it causes at least 6,000 cases of clinical hepatitis each year [2] and many more subclinical cases, infecting 1 in 3,100 blood donors [3]. Those infections are mainly caused by eating pork from infected pigs [4]. In addition, humans can become infected by contact with pigs or their feces, and other environmental transmission routes have been suggested [5]. In most European countries, 50–95% of pigs are being delivered to slaughter with antibodies against HEV and most pig farms are endemically infected [6]. Unfortunately, no commercial HEV vaccine is available for pigs, and due to a limited understanding of how the virus spreads on pig farms, it remains challenging to advise farmers on effective measures to break HEV transmission chains in pig farms.

Pigs become infected with HEV by the uptake of viral particles from feces or urine of other pigs [7]. One infectious pig is estimated to directly infect 4–9 other pigs in a

susceptible population in which all pigs can be in contact with each other [8,9]. Yet, farmed pigs are not all in contact, but housed in separate spaces, namely in pens (a small area to keep pigs of similar age and size), and in turn pens are in compartments (fully confined rooms with multiple pens). In most industrialized countries, pigs that are fattened for human consumption stay in the same compartment for the last four months of their lives (a batch) after which they are transported to slaughter and, following cleaning procedures, a new batch of pigs is housed in the compartment (Fig 1).

Given this separation of pigs in confined spaces, how is it possible that the prevalence of HEV is so high within farms? We hypothesize that environmental transmission, which we define as the indirect spread of a pathogen from one host to another through contact with contaminated surfaces, objects, or elements like water or air, plays an important role. It is considered likely that HEV persists on surfaces and fomites, as the virus can remain infectious in feces and on materials for days to weeks [10], and HEV RNA has been detected in pens after cleaning and disinfection [11]. Between-pen transmission of HEV has been reported by others [8,12], yet the contributions of environmental transmission between farm compartments, and between consecutive batches of pigs, have not been investigated.

Epidemiological field data are required to investigate the contribution of environmental transmission routes to the spread of HEV in pig farms. Experimental data from previous studies are based on conditions such as a low number of pigs per pen and pens spaced much farther apart than usual on farms [8,12], which may result in findings that do not accurately reflect true transmission dynamics. Yet, in the field it is challenging to track specific pig-to-pig contacts, or measure the number of infectious HEV particles in the environment over time. Viral phylogenies built from genetic sequences may overcome that challenge, by providing insights into past transmission events.

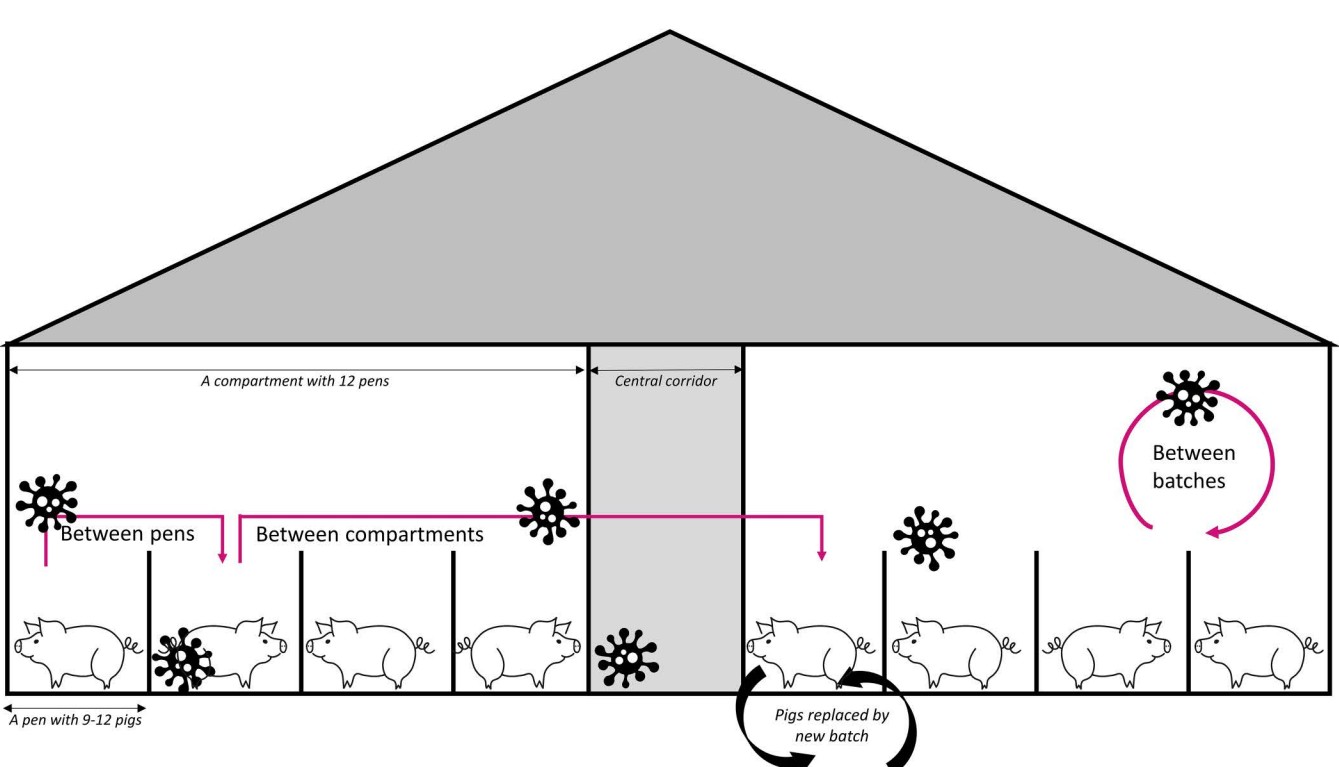

**Fig 1. Schematic overview of a pig farm and the three hypothesized environmental transmission routes (indicated by pink arrows) of hepatitis E virus in farms.**

Here, we integrate epidemiological field and genetic sequencing data in phylodynamic models. We aim to quantify the contribution of environmental transmission routes to the epidemiology of HEV in farms to provide a scientific basis for advising farmers on effective interventions to disrupt HEV transmission chains. Our approach leverages a comprehensive dataset comprising weekly fecal sampling from pens across multiple compartments on two farms for nine months in 2022, coupled with near-complete sequences of HEV from most affected pens. This is the first time that phylodynamic modeling is applied to study viral transmission dynamics of HEV on such a detailed scale.

## Results

### Infection dynamics

To study HEV transmission in the field, we included two Dutch fattening farms (referred to as farm A and farm B), where we conducted weekly fecal sampling in 36 pens across three farm compartments (compartment 1, 2, and 3) during three batches of pigs (A through C) (Fig 2A). The farms were selected because they were found HEV positive, with high (farm A) and low (farm B) seropositivity at slaughter in an earlier study. They are entirely independent, i.e., they are more than 60 km apart, have different veterinarians and buy piglets from different farms. Details about the farms can be found in Text A and Table A in S1 Appendix. The unit of observation was the pen instead of the individual pig, as transmission between individual pigs within pens has been investigated before [12] and measures to prevent spread are hard to conduct between individual pigs inside pens. We call a pen 'infected' when one or more pigs in the pen test PCR positive for HEV, determined by pools of fresh fecal droppings (FD) per pen.

The infection dynamics on the two farms showed a marked contrast (Fig 2B and Tables B and C in S1 Appendix). In farm A, HEV infection occurred in all 36 pens in batch B and C, but was not seen in the final weeks of batch A. Based on HEV PCR positivity, viral shedding started 2–5 weeks after pigs entered the compartments and continued for on average eight weeks (Table D in S1 Appendix). No clear order in the start of shedding per pen could be distinguished. Shedding ended before the pigs were slaughtered (Fig 2B and Table B in S1 Appendix). In farm B, only pigs in compartment 2 became infected in both batch A and B, but not in batch C (Fig 2B and Table C in S1 Appendix). Shedding started in the sixth week of batch B, and a gradient was observed in the onset of shedding, from pens in the right-rear corner to pens closer to the front, until pigs in the pen in the left-front corner started shedding in the week before slaughter. Compartment 1 and 3 remained HEV-negative throughout all three batches (Fig 2B).

Blood was collected from four pigs per pen, at the beginning and end of batch B and C to assess their HEV serological status using an in-house ELISA. Samples with Percentage of Positivity values (PP-values) > 43.5 were considered seropositive. As indicated in Fig 2B, in farm A 30% (43/144) and 18% (26/144) of the sampled pigs tested positive for HEV antibodies at the start of batch B and C respectively, progressing to 93% and 94% of pigs at the end of batch B and C, demonstrating the extent of the outbreak in this farm. In farm B, 3% (4/144) and 6% (9/144) of pigs were seropositive at the start of batch B and C respectively, progressing to 10% at the end of both batches, which were mostly pigs from compartment 2 (14 out of 15 seropositive pigs in batch B). For 98% of pigs from infected pens, PP-values were higher at the end than at the start of the batch. The 14 seropositive samples from farm B batch C had an average PP-value of 69 (cut-off of 43.5), while the seropositive samples from farm A batch C had an average PP-value of 203.

HEV genome sequences were obtained from 30 of 36 infected pens in farm A and 9 of 12 in farm B. Multiple sequences, collected at two to five time points, were obtained from 23 pens (Fig 2B), yielding 89 high-quality genomes with a mean coverage of 3920 reads per nucleotide position, and 7080 bases after aligning and cleaning. Recombination was assessed using RDP4, identifying nine genomes with evidence of either true recombination or mixed signal as samples were pooled (Fig A in S1 Appendix). The putative recombinant regions were masked prior to further analyses. All sequences clustered within HEV genotype 3c, the dominantly detected subtype in The Netherlands [13]. A maximum likelihood (ML) phylogenetic tree was built using the 89 sequences and all European HEV-3c WGSs from database

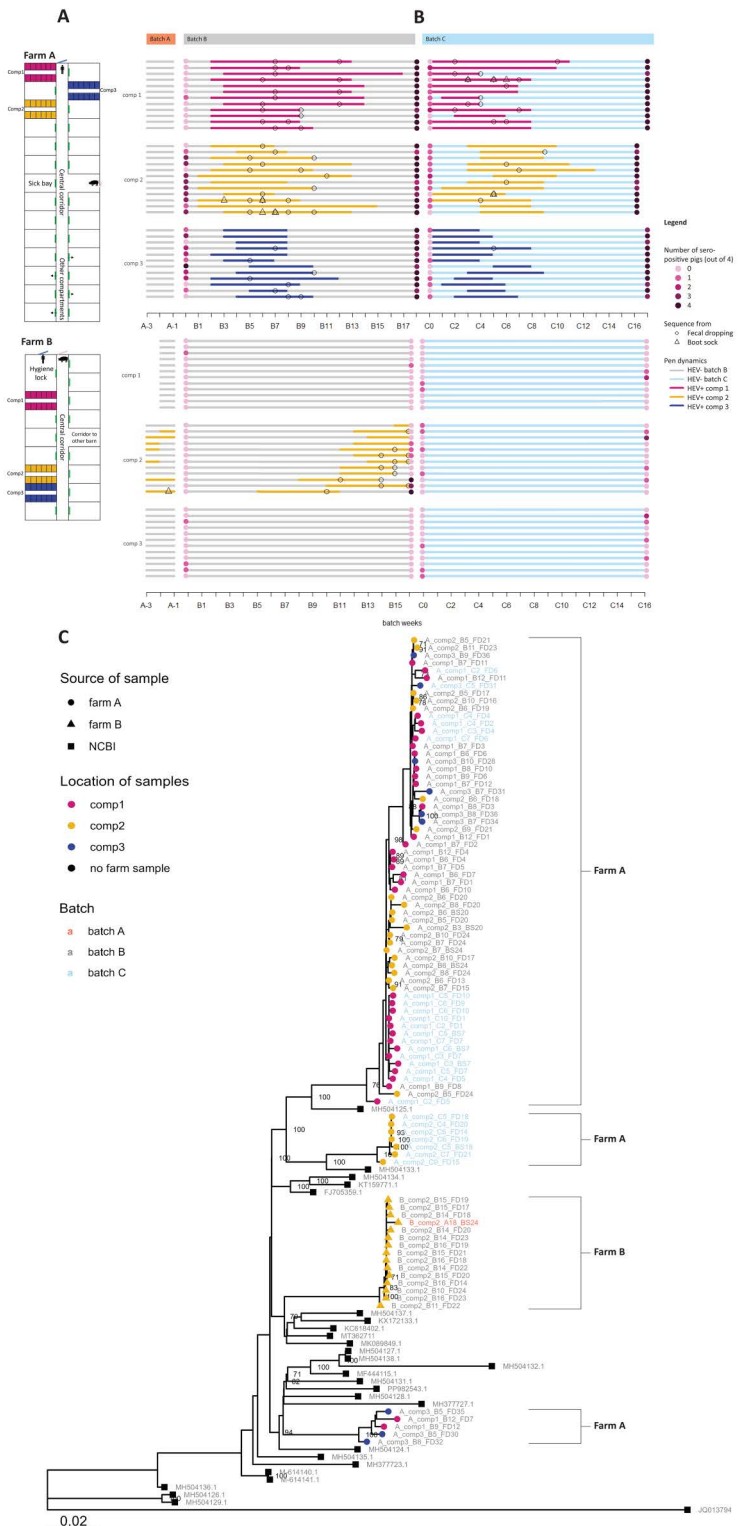

**Fig 2. Infection dynamics and phylogenetics of HEV in two farms. 2A:** Farm structure, indicating all compartments in black and white, and the sampled farm compartments by three colored rectangles, each with twelve pens inside. **2B:** Timeline of the study with three sampled batches and weekly HEV PCR results per pen (horizontal line) per compartment (same colors as 2A). Legend: Serological results at the start and end of batch B and C are

indicated by pink circles that are darker in case more pigs were HEV seropositive (maximally 4). Open circles (fecal dropping samples) and triangles (boot sock samples) indicate weeks in which HEV sequence data was obtained. **2C**: Maximum likelihood tree with all sequences from both farms, and HEV 3c sequences extracted from NCBI. Legend: Tip labels represent the farm (A or **B)**, compartment (1/2/3), batch (A/B/C), week of sampling within the batch, type of sample (fecal dropping (FD) or bootsock (BS) and pen number (1 to 36). Numbers at the internal nodes represent bootstrap support values (>0.70).

NCBI, uploaded until March 2023 and collected between 2006 and 2022 (n = 26, accession numbers in Table E in S1 Appendix) [14]. For pens 20, 24 (batch B) and 7 (batch C) of farm A five or six samples from different weeks of infection were sequenced, to assess whether HEV was introduced in pens multiple times, but all sequences clustered within their respective pens (see tip labels (A_comp2_BX_20, A_comp2_BX_24 and A_comp1_CX_7, with X indicating multiple possible weeks in Fig 2C).

The ML tree shows clustering of sequences on several levels (Fig 2C). Firstly, sequences from farm B formed a separate clade with an average pairwise SNP distance of 15, compatible with a single transmission chain. In contrast, sequences from farm A grouped into at least two clusters, with closely related European sequences, suggesting multiple HEV introductions to the farm. Secondly, there was clustering of sequences per compartment within one batch, suggesting between-pen transmission. Thirdly, for farm A, the phylogenetic ML tree shows high similarity of sequences derived from samples of different farm compartments (e.g., A_comp1_B7_FD11, A_comp2_B5_FD21 and A_comp3_B9_FD36 in top of the tree), pointing towards between-compartment transmission. The results also suggest multiple introductions in compartments of farm A, as sequences obtained from compartments 1 and 3 are represented in different clades that show less similarity to each other than to sequences from other HEV-infected subjects not related to the farm (Fig 2C). Lastly, there was clustering of sequences per compartment over two consecutive batches, suggesting between-batch transmission. These different routes are further quantified with phylodynamic modeling as described next.

### Between-pen transmission

In farm B, the infection dynamics data and phylogenetic tree suggested between-pen transmission after a single introduction. To quantify that transmission, we fitted a Bayesian phylodynamic single-type birth-death (BD) model with sequences from 9 out of 12 pens, to infer the between-pen $R_e$, evolutionary rate and maximum clade credibility (MCC) tree (model specifications in Table F in S1 Appendix). The between-pen $R_e$ is defined as the number of secondary infected pens by one infected pen during its infectious period.

At week 14 of batch B, all pens were HEV PCR positive according to the infection dynamics data (Fig 2B), making transmission between pens after that negligible. Therefore, a parameter-shift was applied in the BD model at week 14, by implementing a change time (Fig 3A, solid vertical line). The median between-pen $R_e$ until week 14 for compartment 2 was estimated at 3.6 (95% highest posterior density interval (HPDI) 1.3-6.7). The MCC tree was consistent with the infection data that suggested the infection started in pen 24, progressing to pen 22 and finally pen 14 (Fig 2B). The estimated onsets of infection of pens according to the BD model (branching times MCC tree) were also comparable to the onsets based on weekly infection data, suggesting that the BD model was adequate in estimating transmission on this scale. The evolutionary rate was estimated at 0.032 nucleotide substitutions per site per year (95% HPDI 0.012-0.063). Other posterior estimates of the BD model are given in Table G in S1 Appendix. To cross-validate the estimates in the phylodynamic model, a Bayesian population dynamic SEIR (Susceptible-Exposed-Infectious-Recovered) model was applied as second approach (Fig B in S1 Appendix). According to this model, the median between-pen $R_e$ until week 14 was 5.7, which is higher than the median $R_e$ of the BD model, although the HPDIs of the models overlap (Fig 3B).

In farm A, pens from multiple compartments and batches became infected so we fitted a multitype birth-death (MTBD) model to estimate the $R_e$ for between-pen and between-compartment transmission. The median estimate for the between-pen $R_e$ of farm A was 1.7 (95% HPDI 0.02–6.3), which is lower than the median estimate of farm B.

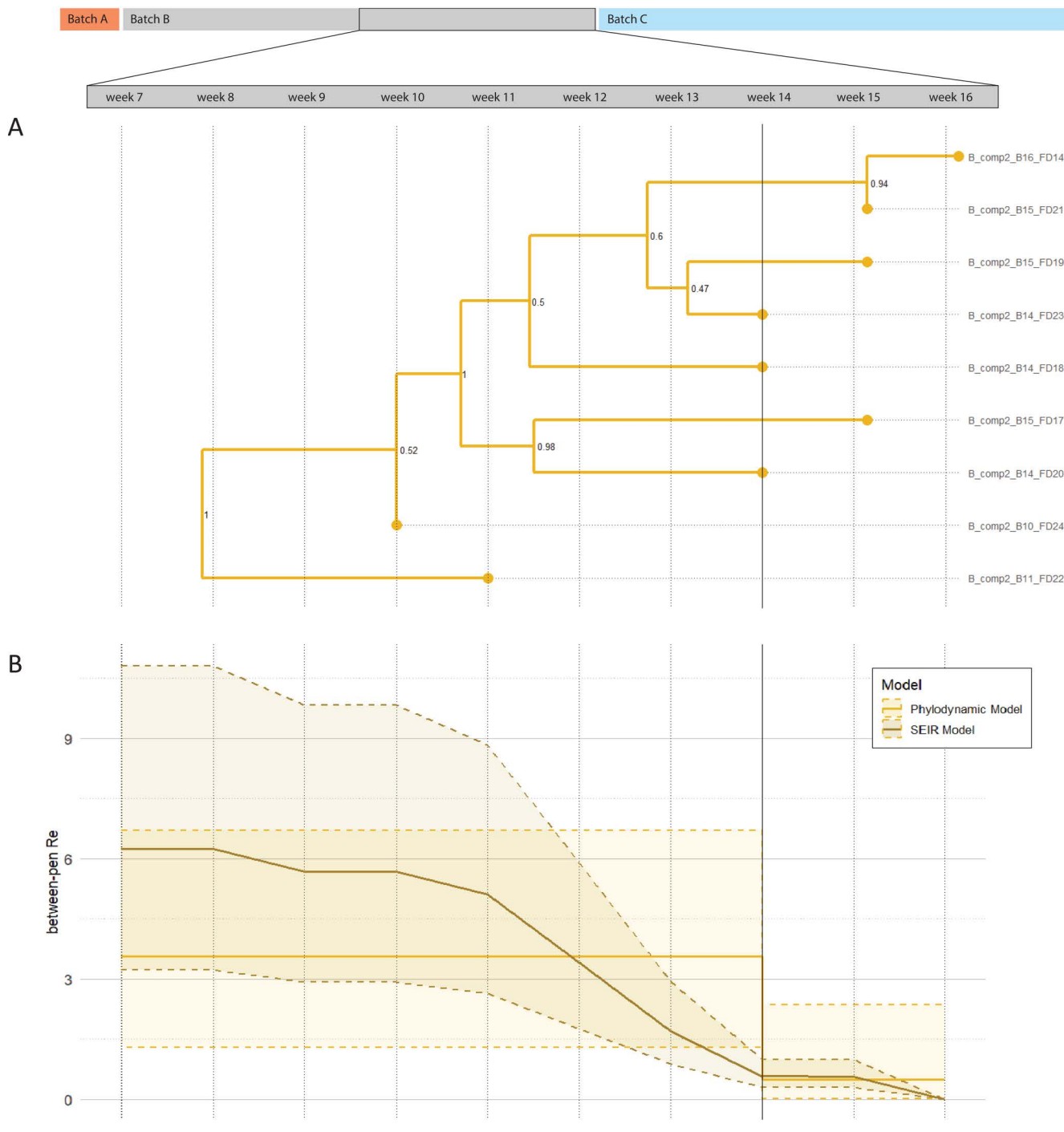

**Fig 3. Output of phylodynamic (BD) and SEIR model to estimate the effective reproduction number of between-pen transmission (between-pen $R_e$) for farm B, compartment 2 during weeks 7 to 16 of batch B.** Legend: The solid line at week 14 indicates the implemented change time for the BD model as all pens were HEV positive in this week according to infection dynamics data. Numbers at the internal nodes represent bootstrap support values. 3A: Maximum clade credibility (MCC) tree of the BD model estimated from 9 whole genome sequences from pens in compartment 2. 3B: Between-pen $R_e$ estimates with 95% highest posterior density intervals from phylodynamic BD and mechanistic SEIR models over time.

## Between-compartment transmission

Between-compartment transmission could only be studied in farm A, as all samples from two out of three sampled compartments on farm B tested negative for HEV. The between-compartment $R_e$ is defined as the number of secondary infected pens in a compartment, caused by one infected pen in another compartment. The MTBD model retrieved a posterior between-compartment $R_e$ of 0.16 (95% HPDI 0.02–0.37), meaning that one infected pen infects a median of 0.16 pens in other compartments, which is ten times lower than between pens within compartments. The MCC tree of the MTBD model had five branching events prior to the start of batch B and an older root than expected for a single transmission chain (Fig 4A).

## Between-batch transmission

For farm A, the low $R_e$ estimates and MCC tree suggest that a significant portion of transmission occurred before batch B started, with several transmission events timed before the pigs were moved to the farm. This is in contrast to the 70% of seronegative pigs, and zero shedding found in the first weeks of batch B. Two scenarios could explain these findings: repeated introductions of independent lineages to the compartment, from outside or from unsampled compartments, or between-batch transmission via persisting strains shed by the previous batch of pigs. The BD model could not be used to assess between-batch transmission due to unsuccessful sequencing for most batch C samples (Fig 2B). Instead, a coalescent Bayesian skyline model was applied on compartment 1 batch B and C sequences, to assess the evolutionary rates of lineages within each batch, and the most recent common ancestor (MRCA) of the two batches (model specifications in Table H S1 Appendix). If between-batch transmission occurred, the most recent common ancestor (MRCA) of batch C sequences should be nested within batch B sequences, and lineages leading to batch C sequences should have lower evolutionary rates, as the virus cannot evolve in the environment.

The retrieved MRCAs of batch C sequences were indeed nested within batch B sequences, and the timing of the MRCA (tMRCA) of clades containing batch C sequences was during HEV shedding in batch B (Fig 4B). The median posterior evolutionary rate estimates of most lineages leading to batch C sequences were lower than those of batch B, yet the HPDIs of the evolutionary rates were overlapping (Table I in S1 Appendix).

Between-batch transmission was also qualitatively assessed by assessing batch clustering in the phylogenetic tree (Fig 2C). For farm A, compartment 1 and 3 sequences clustered regardless of batch of origin, yet sequences from compartment 2 batch C samples were part of a different clade within the tree than those of compartment 2 batch B, suggesting a separate origin. In farm B, the one sequence retrieved from batch A was found in the same clade as the batch B sequences (Fig 2C). Moreover, based on the infection data of farm B, those pens that were infected earliest during batch B, were the same pens in which HEV was shed during the final weeks of batch A, which is consistent with between-batch transmission (Fig 2B).

## Discussion

We combined extensive HEV epidemiological field data with genetic data and identified several routes of environmental transmission in pig farms. The data strongly suggests that HEV is easily transmitted between pens after the virus has entered a farm compartment. This is evidenced by an $R_e$ value for between-pen transmission of above 3 in farm B, and by infected pigs found in each pen of positive compartments in both farms, based on PCR and serology. The high proportion (over 90%) of serologically positive pigs in the week of slaughter is in line with previous research [11,15] and underlines the importance of identifying measures that lower the number of HEV-infected pigs at slaughter.

While between-pen transmission in compartments was efficient in both farms, between-compartment transmission patterns differed. In farm A, multiple genetic sequences clustered across compartments in the phylogenetic tree, and the MTBD model retrieved a low posterior estimate for between-compartment transmission ($R_e$ of 0.16 (95% HPDI 0.02–0.37)), which may imply that between-compartment transmission leads to one or a few introductions per compartment, after which

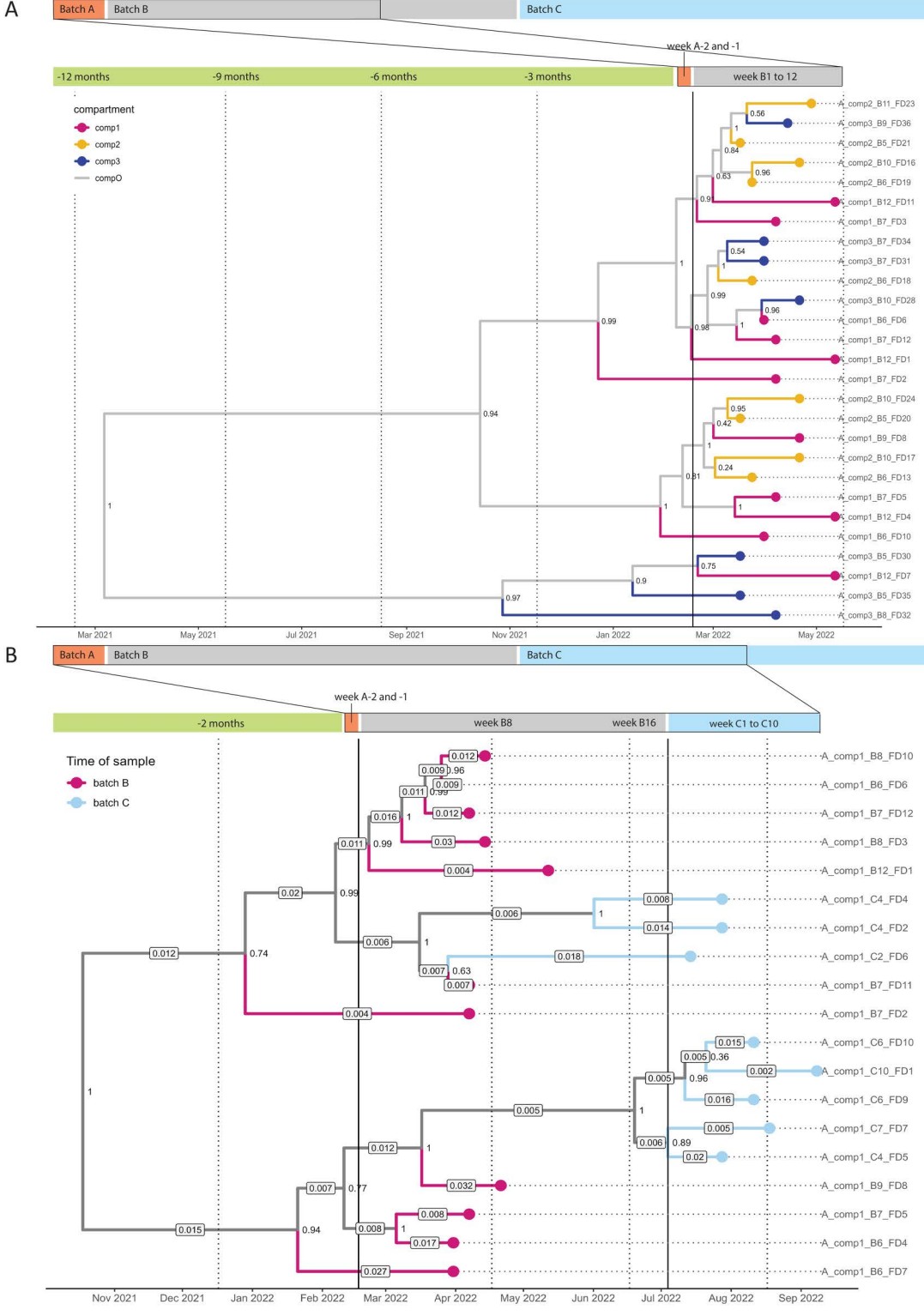

**Fig 4. Maximum clade credibility (MCC) trees of phylodynamic models for transmission in farm A. 4A:** MCC tree of the multitype birth-death model of farm A, batch B, showing clustering of sequences across compartments and branching events prior to the start of the batch **B.** Legend: vertical lines indicate the start of batch B and the implemented moment of a change time. Numbers at the internal nodes represent bootstrap support values.

**4B:** MCC tree of the coalescent Bayesian skyline model of farm A, with sequences from compartment 1, batch B and C, showing nesting of batch C sequences within the batch B sequences. Legend: The branch colors represent the batches. Labels on top of branches represent median posterior evolutionary rate estimates (in substitutions/site/year). The lower evolutionary rates of the branches connecting batch B and C suggest between-batch transmission.

between-pen transmission drives further spread of the introduced strain. Since the $R_e$ for between-compartment transmission reflects the number of pens a single pen can infect outside its compartment, all pens in a compartment together can still well be able to cause sufficient between-compartment transmission for HEV to become endemic.

In contrast to farm A, in farm B no signs of between-compartment transmission were observed. Two of the three sampled compartments remained HEV-negative for over six months, despite twice-daily movement of the farmer from HEV-positive compartment 2, to HEV-negative compartment 3, and the compartments being adjacent to each other (Fig 2A). Sequences from compartment 2 are consistent with a single introduction and subsequent transmission chain. Hypothetically, if farm A's between-compartment $R_e$ would apply to farm B, the 12 positive pens in compartment 2 would have infected a median number of 1.9 pens in other compartments (0.16×12). Given the high $R_e$ of between-pen transmission that would follow after between-compartment transmission events, we would have expected to detect positive pens in other sampled compartments of the farm in case this was true. Nevertheless, we cannot completely exclude between-compartment transmission in farm B, since not all compartments present were sampled. In addition, some pigs from seemingly 'unaffected compartments' tested seropositive at the end of the batches, which could be a signal that some infections were missed, though the average PP-values of positive samples were more than 3 times lower for farm B than A. At the start of the batches, pigs in farm A were more frequently seropositive than in farm B, which may be be due to maternal antibodies, as pigs entering farm A were suspected to be somewhat younger than farm B. Regardless, we can assume that between-compartment transmission was more effective in farm A. Compared to farm A, farm B applied two specific measures that may have contributed to prevention of between-compartment transmission. First, boots were cleaned with high-pressure water and soap before entering compartments, which is a practice associated with a lower HEV farm prevalence [16,17]. Second, the routine checks by the farmer were consistently performed in order from young to old pigs. HEV infection typically occurs during the fattening phase [5], thus the youngest fattening pigs have a lower chance of being HEV-infectious, explaining why the measure could reduce between-compartment transmission. Although these measures are suggested based on two observed farms only, this study provides the first evidence of prevention of HEV transmission to groups of pigs in the field. Therefore, the biosecurity practices of farm B serve as an example of how to tackle between-compartment transmission in endemic farms.

We found clear indications for between-batch transmission in farm A. In farm B there is some evidence pointing to transmission between batch A and B as well, based on sequence similarity. Between-batch transmission can occur when viral particles persist in the environment while remaining infectious until new pigs arrive. Recent studies show that HEV is highly persistent under many conditions, e.g., it is stable against drying on several surfaces and across a range of temperatures, remaining detectable for up to 8 weeks at 3°C [10], it can remain infectious on stainless steel for more than 11 weeks [18], and disinfection with alcohol cannot completely inactivate HEV particles [19]. Pig farmers aim to prevent between-batch transmission through cleaning and disinfection. In farm A, compartments were solely cleaned with high-pressure cold water, leaving visible fecal residues on pen walls and below feed troughs. In farm B, pens were more thoroughly cleaned, involving soaking with water and soap, high-pressure cleaning, and disinfection with a product containing an aldehyde- and a quaternary ammonium-based ingredient (glutaraldehyde, 14.7% and alkyl-(C12-16)-dimethylbenzylammoniumchloride, 9.8%), which were both shown to be effective in inactivating HEV genotype 3 [18]. The differences in cleaning procedures may explain less signal for between-batch transmission in farm B. However, the potential between-batch transmission observed between batch A and B in farm B, suggests that complete HEV inactivation in a farm compartment is challenging, even with a thorough cleaning protocol.

Our approach to apply phylodynamic models to a small outbreak in a highly structured population, is unique. The models are usually applied to large outbreaks where the proportion of cases being sequenced is low, and the true timing of infection unknown. In the current study, a sample from most cases (pens) was sequenced, and the timing and duration of infection were known thanks to longitudinal sampling. The Bayesian inference method, allowed us to inform the phylodynamic model with the infection data and reduce the number of parameters to be estimated. The evolutionary rate of HEV had to be informed from prior knowledge, because the sequences were collected over a too short period to estimate an evolutionary clock rate directly. The issue was solved by running several models on farm B data with different times of the first transmission event (MRCA), based on the longitudinal results, and combining the posterior evolutionary rates to an informed uniform prior in the final model of farm A. Although the estimated evolutionary rate was about ten times higher than previously reported [20–22], it falls within the 95% HPDI estimated by Pisano et al. [23]. The elevated evolutionary rate could be attributed to incomplete purifying selection due to the short sampling interval [24]. Future outbreak studies with a short timespan should consider that published evolutionary rates may not apply to the outbreak.

To cross-validate the BD model, we compared its $R_e$ estimates for between-pen transmission with those from a mechanistic SEIR model, commonly used in infectious disease transmission studies [12]. The SEIR model yielded a higher median $R_e$, which may be due to multiple factors. Firstly, the SEIR model included a one-week latent period ('E') as latent periods have been described for HEV infection in pigs [7]. This could not be included in the BD models without overparameterization, thereby underestimating the $R_e$. Removing the 'E' from the SEIR reduced the $R_e$ from 6.3 to 5.3 in the first week, aligning more closely with the BD model. Secondly, the MTBD model assumes an infinite number of susceptible pens, whereas in a compartment with twelve pens, susceptibles deplete quickly. Although we introduced a change time at the moment that all pens were infected, susceptible depletion is a continuous process and excluding that may have led to further underestimating $R_e$ values for both farms. Still, the HPDIs of $R_e$ estimates in both models largely overlap (Fig 3B). While SEIR models have the advantage of solely requiring case count data in time, and being computationally efficient, phylodynamic models may better distinguish different routes of (environmental) transmission in structured populations, especially in situations where multiple strains circulate (farm A). Future phylodynamic models should account for periods without evolution, or variable evolutionary rates due to persistence in the environment or infection of alternate hosts, to further elicit environmental transmission routes of pathogens.

In conclusion, this study disentangled the contribution of environmental transmission to HEV infection dynamics in farms. By innovative phylodynamic modeling, we have observed that while HEV transmission between farm compartments and batches occurs, it can be prevented, whereas between-pen transmission appears inevitable. We recommend farmers to improve biosecurity measures to reduce spread between compartments, for example, cleaning of boots or tools used across compartments. Moreover, cleaning and disinfection procedures must be critically evaluated to ensure HEV inactivation before new pigs arrive. Research that quantifies the effects of cleaning methods and other biosecurity measures on HEV dynamics is needed to provide evidence-based advice about HEV control. Ultimately, the current study contributes to the future reduction of HEV-infected pigs, thereby reducing human exposure to HEV.

## Methods

### Ethics statement

This study was conducted in accordance with the guidelines of Good Experimental Practices (GEP) standardly dictated by the European Union, Directive 2010/63/EU. The project was approved by the Dutch Central Committee of Animal Experimentation under license number AVD1080020197664 and the experiment (protocol number: 7664-1-01) was approved by the Animal Welfare Body Utrecht.

## Farm selection

Two conventional fattening pig farms with different infection dynamics of HEV were selected, based on previous research [25]. In that study, pigs from farm A were positive for HEV antibodies during every delivery to slaughter, with 77% of pigs being seropositive across batches, whereas deliveries of pigs from farm B were sometimes seronegative and PCR negative in blood, and in total only 40% were seropositive [25]. Farm characteristics are described in Text A in S1 Appendix.

## Data collection

The cohort study was conducted from January to November 2022. In both farms, three consecutive batches of fattening pigs were followed. Sampling started one to three weeks before the slaughter of the first batch of pigs (batch A) and continued for two consecutive batches from the start of the fattening phase until slaughter (batches B and C). This resulted in a 39-week period for farm A and a 38-week period for farm B (Fig C in S1 Appendix). Fattening pig farms usually house several age groups of pigs in different compartments, i.e., there is a periodical influx and outflux of a part of the pigs in the farm. The specific compartments 1, 2, and 3 (Fig 2A) were selected because these pigs had the right age (two weeks before slaughter age) when data collection started. The preparation for sampling and measures taken to prevent fomite transmission via the researchers during sampling, are described in Text A in S1 Appendix.

Pooled fecal dropping (FD) samples were collected weekly from each pen to determine the duration of the HEV infectious period of pigs at the pen level. An FD sample consists of scoops of about 0.2 grams of feces from multiple fresh droppings inside a pen. The number of droppings sampled per pen was determined by sample size calculations for 'presence of disease'. With a population size of 12 pigs per pen in farm A and 9 in farm B, a minimum proportion of 0.25 affected pigs and a desired probability of 95% of detecting presence of disease, the calculation resulted in 8 and 7 droppings per pen for farm A and B respectively [26].

In addition, boot sock (BS) samples were collected, as earlier research demonstrated that BS have a high sensitivity in case of a low pen-prevalence, resulting in earlier detection of infection in pens [27]. BS samples were therefore mainly collected in the first few weeks of the batches (see Fig C and Tables B and C in S1 Appendix for the exact sampling protocol). The FD and BS results were combined to derive the onset of infection and infectious period per pen according to the method described in Text A in S1 Appendix.

Finally, blood samples were collected to determine HEV serological status of four pigs at the start and end of batch B and C. Details of the sampling kit preparation, sample collection and laboratory protocols to analyze the samples, are described in Text A in S1 Appendix.

FD samples from the first and last week of HEV PCR positivity with a Ct-value <30 were sequenced from each pen and batch. In case a pen solely had FD samples with a CT-value >30, BS samples were selected for sequencing. To assess the genetic diversity of HEV within a pen (i.e., multiple introductions per pen or evolution over time in a pen), 19 samples (both BS and FD) were selected from three pens in farm A instead of the standard two samples per pen. All selected samples were sequenced according to the protocol in Text A in S1 Appendix.

## Phylogenetic analyses

After multiple sequence alignment with MAFFT v. 7 [28] and visual inspection with AliView v. 1.28 [29], nucleotide positions with less than 50% coverage across all sequences (127 positions, at beginning and end of sequences) and sequences with less than 90% coverage were removed (N = 40, original base lengths ranged from 6459 to 7206). Sequences were typed with the Hepatitis E Virus Genotyping Tool v. 1.0 [30], based on reference sequences from Smith et al. [31].

The sequences were subsequently analyzed with Recombination Detection Program 4 (RDP4) [32] using the RDP, GENECONV, BootScan, MaxChi, Chimaera, SiScan, and 3Seq detection methods. Nine sequences that were detected as recombinant in four or more tests, were manually checked for recombination. All nine sequences were from farm A, and

the major and minor parents were always from farm A as well, or 'unknown' (Fig A in S1 Appendix). Because the number of sequences per pen was limited, the putative recombinant regions were masked in all subsequent analyses. One sequence in which more than 50% of nucleotides had to be masked was excluded from subsequent analyses.

Twenty-six European HEV genotype 3c sequences were obtained from NCBI and used to build the phylogenetic ML tree using RAxML-NG (v. 1.1) with an HKY$+\Gamma_4$ substitution model and 1000 bootstrap replicates [33]. A root-to-tip analysis was performed using TempEst (v.1.5.3) to explore the temporal signal of the sequences [34]. The ML trees were visualized using R, with packages ggtree, treeio, ape and tidytree [35–38].

## Phylodynamic analyses

**Multitype birth-death analysis.** We quantified population dynamics of HEV with an MTBD model, as described by Kühnert et al. [39].

The MTBD model is a phylodynamic model that can estimate effective reproduction numbers for compartmentalized populations, such as pig farms, while accounting for sampling biases and temporal changes in transmission dynamics [39]. The model assumes an initial single pen with infected pigs in one subpopulation, followed by a transmission event (birth, without latent period), recovery from infection (death), or sampling in one of the subpopulations, while allowing these parameters to change in a piecewise constant fashion and be different for each subpopulation [40]. The dynamics were reconstructed using the BDMM-Prime and Feast packages in BEAST v2.6 [41–43] in a Bayesian framework to account for uncertainty in the phylogenetic tree [40].

For farm B, the MTBD model consisted of one compartment, as no evidence of HEV infection was found in the other two compartments, i.e., all samples tested negative for HEV. The between-pen $R_e$ was given a lognormal(2.0, 1.25) prior distribution corresponding to a median $R_e$ of 0.91 (95% percentile range (PR) 0.079 – 7.2). There was no temporal signal in the sequence data to estimate an evolutionary rate. Yet, the clock rate could be inferred using longitudinal infection data from farm B, by information about the timing of first transmission (tMRCA) and the timing of sample collection. This indicated that the first transmission event between pens in comp 2 most likely occurred around week 8 of batch B (shedding was detected in week 9 in the second pen, we subtracted 1 week to account for the latent period of HEV infection, also see Table C in S1 Appendix), and no earlier than week 2 (very conservative as shedding was only detected in week 6 in the first pen) of batch B (Fig 2B). Thus the model for farm B was run twice, with a strong prior for the tMRCA with the mean set to week 2 and week 8 respectively (Laplace distribution, μ at week 2 and 8, scale at 0.0003), and using an uncorrelated relaxed clock (with a logNormal(0.001, 1.25) prior for the mean, and gamma(0.537, 0.382) prior for the standard deviation) to estimate the clock rates. The uncorrelated relaxed logNormal clock rate (UCLD) is justified, as the HPDI of the posterior coefficient of variation among branches did not include 0 in the farm B models. In the results section the outcomes of the model with week 8 as prior for the tMRCA are reported as week 8 is the most realistic timing of first transmission. The model outcomes with week 2 are in the same order of magnitude as the week 8 model with regards to the $R_e$ and evolutionary rate (shown in Table G in S1 Appendix).

For farm A, the MTBD model included compartment 1–3, and 'compO'. CompO represented the pens in all unsampled compartments of the farm (Figs 2A and D in S1 Appendix), because those pens may also have contributed to infection in the sampled pens. A separate $R_e$ for transmission from compartment 1/2/3 to compO was modeled because the infection dynamics in compO are a mixture of between-pen and between-compartment transmission and compO contains a larger number of susceptible pens. The between-comp $R_e$ and the compartment 1/2/3 to compO $R_e$ in the farm A MTBD model were given logNormal(1.0, 1.25) prior distributions (median 0.46, 95% PR 0.040 – 5.3) (motivation for choice of priors in Table F in S1 Appendix). As in farm B, there was no temporal signal for the farm A sequence data. However, it was not possible to bound the clock rate using the time of the first transmission, since there were multiple introductions to the farm. Instead, the upper and lower bounds of the 95% HPDIs of the mean clock rates from the two farm B models (Table F in S1 Appendix) were used to bound the clock rate with a uniform prior (0.0124 ≤ mean clock rate ≤ 0.0408). Priors for

the other parameters in the MTBD model are provided in Table F in S1 Appendix. A UCLD was then applied, while also estimating the tMRCA, effectively marginalising over the clock rate uncertainty observed in farm A. A change time was introduced in the model at the start date of batch B, to model $R_e$ values for transmission during batch B, without incorporating transmission prior to the start of the batch. The sampling proportion of the compartments was fixed to 0 before the change time (Table F in S1 Appendix).

For both farms, one FD sample sequence per pen from the earliest week of batch B was included in the MTBD model, although additional models with a random selection of an FD or BS sample sequence per pen were run to test robustness of the results. An HKY$+\Gamma_4$ substitution model was used [44]. The rate of recovery of infection was fixed at 6.5 per year, based on the number of weeks per year (52) divided by the duration of the infectious period according to the infection data (8 weeks). Sampling proportions were fixed to the number of available sequences divided by the number of infected pens. The $R_e$ may be underestimated because a quick depletion of the susceptible population occurs in farms, which cannot be taken into account in MTBD models [39]. To mitigate this issue, separate $R_e$ values were estimated for the period until all pens per compartment were HEV positive, and the period after that, by applying an additional change time (Fig E in S1 Appendix). The MTBD models were run with three chains of 100,000,000 Markov-chain Monte Carlo (MCMC) steps of which every 5000th step was sampled for posterior estimates of the models.

**Coalescent Bayesian skyline analysis.** Potential between-batch transmission, the evolutionary rate of batch B and C sequences, and nesting of the MRCAs of 19 batch B and C sequences were assessed using a Bayesian coalescent skyline model, with 6 sequences also used in farm A's MTBD model. In a coalescent Bayesian skyline model past population dynamics are inferred through time by assessing the time needed for two taxa to coalesce [45]. The model was run in BEAST v2.6 [45], with five time segments and an HKY$+\Gamma_4$ substitution model. A UCLD model was chosen to allow clock rates to vary among branches. For the mean clock rate the same prior was used as in the MTBD model for farm A (Uniform(0.0124 ≤ mean clock rate ≤ 0.0408)). A gamma(0.537, 0.382) prior was applied for the standard deviation of the relaxed clock. The chain length of the coalescent Bayesian skyline model was set at 10,000,000 steps and every 1000th step was logged.

For all phylodynamics analyses, maximum clade credibility (MCC) trees were built from three separate chains in TreeAnnotator using median node heights [46]. A burn-in of 10% was excluded from all model outputs. Convergence was assessed by visual inspection of trace files, and an effective sample size of >200 was reached for all parameters.

**Mechanistic SEIR model.** The pens were classified 'I' following the rules about determining onset of infection as described in Text A in S1 Appendix. One week prior to the onset of infection, the pens were classified 'E', assuming a latent period of one week [6]. The pens were classified 'S' the weeks prior to that, and 'R' after the infectious period had passed. Herewith the number of pens with infectious or susceptible pigs at the start of each time interval ($\Delta t$ = one week) and new cases ('C') during the interval could be determined. The expected number of cases per week $\mathbb{E}[C]$ was modeled as described by [47]:

$$\mathbb{E}[C] = S\left(1 - e^{-R_0 \times \gamma \times I \times \Delta t}\right)$$

(1)

with $R_0$ representing the reproduction number between pens and $\gamma$ the rate of recovery (Fig B in S1 Appendix). The rate of recovery was inferred from the weekly infection data, as 1 divided by the infectious period on pen-level. This was modelled in a Bayesian generalized linear model with complementary log-log link function, and $log(\gamma \times I \times \Delta t)$ as offset, giving the following statistical model:

$$cloglog\left(\frac{\mathbb{E}[C]}{S}\right) = \log(R_0) + log(\gamma \times I \times \Delta t)$$

(2)

The median $R_0$ and 95% HPDI were determined by exponentiation of the intercept and 5% and 95% percentile of the posterior distribution of $\log(R_0)$ was multiplied by the fraction of susceptible pens (Fig 2) in each week to determine $R_e$.

## Supporting information

**S1 Appendix.** **Text A.** Farm description. Preparation of sampling kits for sample collection. Collection of boot sock and blood samples. Prevention of HEV transmission via researchers. Laboratory protocol. Criteria to define the onset of infection and infectious period on pen-level. HEV sequence protocol and alignment. **Table A.** Farm characteristics, production parameters and biosecurity measures. **Table B.** Ct-values of boot sock (BS) and fecal dropping (FD) samples of farm A, per pen, for each week of sampling. **Table C.** Ct-values of boot sock (BS) and fecal dropping (FD) samples of farm B, per pen, for each week of sampling. **Table D.** Time of onset and duration of pen-level infection per farm compartment and batch. **Table E.** List of HEV genotype 3c complete genome sequences retrieved from NCBI used for the phylogenetic maximum likelihood tree. **Table F.** Model specification and priors for the multitype birth-death models of farm A and B. **Table G.** Results of MTBD models of farm A and farm B. **Table H.** Model specification and priors for the coalescent skyline analysis of farm A. **Table I.** Estimated posterior evolutionary rate distributions of batch B and C sequences in the coalescent Bayesian skyline analysis. **Fig A.** Suggested parent strains and regions of recombination of nine recombinant sequences from farm A. **Fig B.** Schematical overview of the models for between-pen transmission of compartment 2 in farm B. **Fig C.** Sampling scheme of fecal dropping, boot sock and blood samples per farm. **Fig D.** Schematical overview of multitype birth-death model for farm A, batch B. **Fig E.** Change times and parameters separately estimated before vs. after change times.
(DOCX)

## Acknowledgments

The authors would like to thank the farmers for welcoming the researchers on their farm and participating in the practical work. Moreover, the authors are grateful for all students who participated in the collection of samples and laboratory work. Lastly, the colleagues from the Computational Evolution group of ETH Zürich are thanked for advising and helping the first author in setting up the phylodynamic analyses.

## Author contributions

**Conceptualization:** Marina Meester, Tijs J. Tobias, Martijn Bouwknegt, Wim H.M. Van der Poel, Arjan Stegeman.

**Data curation:** Marina Meester.

**Formal analysis:** Marina Meester, Cecilia Valenzuela Agüí, Claire Guinat.

**Funding acquisition:** Tijs J. Tobias, Wim H.M. Van der Poel, Arjan Stegeman.

**Investigation:** Marina Meester, Tijs J. Tobias, Martijn Bouwknegt.

**Methodology:** Cecilia Valenzuela Agüí, Renate W. Hakze - van der Honing, Claire Guinat, Louis Du Plessis, Egil A.J. Fischer, Tanja Stadler.

**Project administration:** Tijs J. Tobias.

**Resources:** Renate W. Hakze - van der Honing, Mirlin P. Spaninks, Tanja Stadler.

**Supervision:** Tijs J. Tobias, Claire Guinat, Martijn Bouwknegt, Tanja Stadler, Wim H.M. Van der Poel, Arjan Stegeman.

**Visualization:** Marina Meester.

**Writing – original draft:** Marina Meester.

**Writing – review & editing:** Cecilia Valenzuela Agüí, Tijs J. Tobias, Renate W. Hakze - van der Honing, Claire Guinat, Martijn Bouwknegt, Louis Du Plessis, Egil A.J. Fischer, Tanja Stadler, Wim H.M. Van der Poel, Arjan Stegeman.

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
