## [Decision Letter · Decision Letter 0]

28 Aug 2025

Zoonotic hepatitis E virus spreads through environmental routes in pig herds - a phylodynamic analysis

PLOS Pathogens

Dear Dr. Meester,

Thank you for submitting your manuscript to PLOS Pathogens. After careful consideration, we feel that it has merit but does not fully meet PLOS Pathogens's publication criteria as it currently stands. Therefore, we invite you to submit a revised version of the manuscript that addresses the points raised during the review process.

Please submit your revised manuscript within 60 days Oct 27 2025 11:59PM. If you will need more time than this to complete your revisions, please reply to this message or contact the journal office at plospathogens@plos.org. Please include the following items when submitting your revised manuscript:

We look forward to receiving your revised manuscript.

Kind regards,

Daniel Todt

Guest Editor

PLOS Pathogens

Ronald Swanstrom

Section Editor

PLOS Pathogens

Editor-in-Chief

PLOS Pathogens

orcid.org/0000-0003-2946-9497

Editor-in-Chief

PLOS Pathogens

orcid.org/0000-0002-7699-2064

**Additional Editor Comments:**

As you will find in the comments, the reviewers had opposing opinions on the manuscript. During the revision process, please focus on the issues raised by reviewer #3, especially regading the phylogenetic analysis, e.g. masking segments with recombinant signal and recombination analysis. The other reviewers were quite in favor and made useful suggestions to improve the clarity of your manuscript.

**Journal Requirements:**

At this stage, the following Authors/Authors require contributions: Marina Meester, Cecilia Valenzuela Agüí, Tijs J. Tobias, Renate W. Hakze - van der Honing, Claire Guinat, Martijn Bouwknegt, Louis Du Plessis, Egil A.J. Fischer, Mirlin P. Spaninks, Tanja Stadler, Wim H.M. Van der Poel, and Arjan Stegeman. Please ensure that the full contributions of each author are acknowledged in the "Add/Edit/Remove Authors" section of our submission form.

Potential Copyright Issues:

i) Figures 1, 2A, and 2B. Please confirm whether you drew the images / clip-art within the figure panels by hand. If you did not draw the images, please provide (a) a link to the source of the images or icons and their license / terms of use; or (b) written permission from the copyright holder to publish the images or icons under our CC BY 4.0 license. Alternatively, you may replace the images with open source alternatives. See these open source resources you may use to replace images / clip-art:

7)  Please ensure that the funders and grant numbers match between the Financial Disclosure field and the Funding Information tab in your submission form. Note that the funders must be provided in the same order in both places as well.  

**Reviewers' Comments:**

Reviewer's Responses to Questions

**Part I - Summary**

Reviewer #1: Meester and colleagues present a comprehensive study of the transmission dynamics of the HEV on pig farms. This research is important for understanding HEV outbreak scenarios, with significant implications for improving outbreak management and overall public health. The authors used sophisticated phylogenetic models to shed light on how HEV is transmitted within farm settings, particularly between pens and compartments. The study is well written and thoroughly explained, and it is of importance for human and animal safety.

I have a few minor comments and suggestions:

Reviewer #2: The article “Zoonotic hepatitis E virus spreads through environmental routes in pig herds – a phylodynamic analysis” by Meester et al describes a well-done epidemiological surveillance study of HEV environmental transmission in two pig farms. The authors were able to track HEV transmission and dynamics in three pig batches through phylogenetic means, and determine the effective reproduction number. The authors found differences in transmission dynamics between both farms, likely due to the biosecurity measures taken.

Reviewer #3: In this study, epidemiological field data and genetic sequencing data are integrated into phylodynamic models. The objective of this study is to quantify the contribution of environmental transmission routes to the epidemiology of HEV in agricultural holdings. This will provide a scientific basis for advising farmers and other stakeholders on effective measures to interrupt HEV transmission chains in pig production.

The present study is founded upon a comprehensive dataset, comprising weekly faecal samples from stables in several compartments on two farms over a period of eight months, coupled with whole genome sequencing of HEV. This is the first instance of phylodynamic models being employed to study the virus transmission dynamics of HEV on such a detailed scale.

This manuscript presents a clear, well-designed, and highly relevant study that makes a valuable contribution to the field. The research question is articulated with clarity, the methodology is rigorous, and the findings are presented and interpreted in a convincing and excellent manner. The work is clearly written and well structured, and will be of significant interest to the journal's readership.

Reviewer #4: This is a good study performing detailed high density sampling and sequencing, and advanced phylodynamic analysis of hepatitis E virus in two commercial pig farms in the Netherlands. The whole genome sequences in this study were collected within a year, and were used to show introductions on farms, spread within the farm and between batches of animals, and the difference that on-farm biosecurity measures can make to control the virus.

**Part II – Major Issues: Key Experiments Required for Acceptance**

Reviewer #1: (No Response)

Reviewer #2: The study is a well-done study with clear public health message. However, it does not seem to fit the scope of PLOS Pathogens, it rather fits more for an epidemiologic surveillance type of journal. Environmental transmission routes of HEV is not novel, even if this is the first study assessing its importance in pig farms in which pigs are in separated compartments. The study could be more concisely described, at times it is unclear what the authors mean, and it can likely be shortened, particularly the discussion.

Reviewer #3: none

Reviewer #4: (No Response)

**Part III – Minor Issues: Editorial and Data Presentation Modifications**

Reviewer #1: It would be beneficial for readers to know whether HEV acquired amino acid substitutions during the study. Visualising these substitutions could provide deeper insights into the virus evolutionary dynamics within the farm environment.

It would also be interesting to explore whether there are specific genomic regions that are more prone to mutations in this setup. Identifying such regions could improve our understanding of HEV adaptability and transmission mechanisms.

The authors have discussed the role of hygiene measures in HEV transmission. however, what about viral load in the animals? Has this been quantified? And do the authors think that this could also influence transissmion dynamics?

Reviewer #2: - Some statements need to be softened, as per example HEV is not the primary cause of viral hepatitis worldwide, it is “a” main cause. Perhaps the authors meant “acute viral hepatitis”?

- Yes, HEV has a short genome, but it is not as small as HBV, perhaps take out “small” in Line 61.

- Also, please state that it 1 in 8 people having been infected is just an estimate.

- The authors also point out that human infection of HEV is mainly caused by consumption of raw or undercooked port, and that this route of infection can be controlled by preventing HEV spreading between pigs. I would suppose that more than the consumption of raw or undercooked meat, human infections due to contact with infected pigs at pig farms or slaughters is also high. Consumption of raw or undercooked meat can also mean that the message is to cook well the meat before consumption.

- The authors could state what is the status on the development of vaccine and other treatments.

- Many major clades in the ML tree in Fig 2C has low support. Either include in your discussion or try another method (perhaps Bayesian).

- Be wary on using complete genomes for your phylogenetic analyses, I would suggest to also do the analysis taking out segments with recombinant signal

- The authors state the no temporal signal was found and therefore no evolutionary rate could be detected this way. I would suggest doing a recombination analysis, and see if by removing those regions with recombinant signal, you will get temporal signal to estimate the evolutionary rate and compare to the one you inferred by using longitudinal infection data.

- The figure legends lack detail and should be below the figures as instead of somewhere in the middle of the manuscript to avoid going up and down in the document.

- Please include a scale bar in all your phylogenetic trees

- I suppose the number at the nodes of the trees is support value?

- The authors could consider updating their dataset from March 2023, more than two years ago.

Reviewer #3: Only minor revisions are required prior to publication, as outlined in the uploaded commented PDF file in the Reviewer Attachment (please refer to the 25 comments in the PDF file). These points do not detract from the overall quality of the manuscript, but rather serve to further enhance it.

Reviewer #4: I have a number of suggestions and comments which will hopefully improve the clarity of the work:

Please add in the abstract / author summary and introduction the country and year where the sampled farms are located (this will greatly assist farmers veterinarians and policy makers to be able to find your study and quickly understand if its conclusions are applicable to their situation).

Results

Since the Methods come at the end of the paper, please add a few more basic details into the early part of the Results section.

line 106 - 109. Please add the year of the sampling (2022).

Also, are the farms independent of each other ? e.g. they are >10km apart ?

line 139 - 140. "yielding 89 high-quality genomes", please can you indicate the length in base pairs (and/or coverage).

line 143 "all European HEV-3c WGS" - Please include the date range of the NCBI sequences in the main text as well as referring the reader to the supplementary table (e.g. 2006-2017). Also what was the length of these ? (the exact same as the new sequences, or are some just 90% of whole genome etc ?); this information can be useful for understanding how reliable the ML tree is (in general terms).

Line 150 - "separate clade" - presumably this has a bootstrap value of 100%, can you indicate that on the figure (there are selected bootstrap values on the figure just not in the clade you are talking about).

line 152 (figure 2) "closely related European sequences" - what are the country and year of those sequences ? impossible to tell from the figure 2C. The point being that if they are not from the Netherlands or 2022, then that is supporting the idea that these are different and independent introductions onto the farms.

Figure 2C - taxa labels; for the main manuscript, I agree that you do not really want to put all the (non farm) sequence names on and need to use colours and shapes to distinguish your samples. However, the Farm A or B, and Batch A, B, C are still quite confusing. Suggest that you also put vertical bars on the figure indicating which clades are Farm A and Farm B.

Line 167 - Farm B, please add the number of sequences that went into the birth-death tree.

Line 232 - Comment on Figure 2C; there are only two sequences in the supplementary that dont have dates - you might be able to run a time-scaled tree including the NCBI sequences which do have dates (and check in the records for more details than only the year) ? this would establish a longer term clock rate and allow approximate estimates of the TMRCA of the independent introduction clades, prior to doing the more detailed analyses.

Figure 4A and 4B - is there overlap between the sequences in 4A and those in 4B - if so please can you indicate which ones on the figure. Also, for Figure 4A, which is the multi-type birth death model of farm A, if you removed the bottom 4 sequences which look to be from a separate introduction, how does this affect the Re and evolutionary rate estimates ? the long connecting branches between the two clades could introduce extra uncertainty.

Discussion

figure 2A; in Farm A you show that there are more seropositive pigs at the start of the batch than in farm B. At what age (roughly) do the pigs arrive, and is there any pig age effect in their ability to show a seropositive status if infected ? perhaps include a brief statement in the discussion ? Also in figure 4A, the TMRCAs within the upper clade are a few months prior to arrival; how do these relate to the age of the pigs ?

PLOS authors have the option to publish the peer review history of their article (what does this mean? ). If published, this will include your full peer review and any attached files.

**Do you want your identity to be public for this peer review?** For information about this choice, including consent withdrawal, please see our Privacy Policy .

Reviewer #1: No

Reviewer #2: No

Reviewer #3: No

Reviewer #4: No

**Figure resubmission:**

**Reproducibility:**



---

## [Editor Report · Decision Letter 1]

11 Nov 2025

Dear Dr. Meester,

We are pleased to inform you that your manuscript 'Zoonotic hepatitis E virus spreads through environmental routes in pig herds - a phylodynamic analysis' has been provisionally accepted for publication in PLOS Pathogens.

Best regards,

Daniel Todt

Guest Editor

PLOS Pathogens

Ronald Swanstrom

Section Editor

PLOS Pathogens

Sumita Bhaduri-McIntosh

Editor-in-Chief

PLOS Pathogens

orcid.org/0000-0003-2946-9497

Michael Malim

Editor-in-Chief

PLOS Pathogens

orcid.org/0000-0002-7699-2064

The authors addressed all issued raised by the reviewer in a satisfactory manner.

Congratulaitons on this interesting study!
---

## [Editor Report · Acceptance letter]

Dear Dr. Meester,

We are delighted to inform you that your manuscript, "Zoonotic hepatitis E virus spreads through environmental routes in pig herds - a phylodynamic analysis," has been formally accepted for publication in PLOS Pathogens.

Best regards,

Sumita Bhaduri-McIntosh

Editor-in-Chief

PLOS Pathogens

orcid.org/0000-0003-2946-9497

Michael Malim

Editor-in-Chief

PLOS Pathogens

orcid.org/0000-0002-7699-2064